# PC²: Pseudo-Classification Based Pseudo-Captioning for Noisy Correspondence Learning in Cross-Modal Retrieval

**Yue Duan**[*]
Nanjing University
Nanjing, China
Ant Group
Shanghai, China
yueduan@smail.nju.edu.cn

**Zhangxuan Gu**
Ant Group
Shanghai, China
guzhangxuan.gzx@antgroup.com

**Zhenzhe Ying**
Ant Group
Hangzhou, China
zhenzhe.yzz@antgroup.com

**Lei Qi**
Southeast University
Nanjing, China
qilei@seu.edu.cn

**Changhua Meng**
Ant Group
Hangzhou, China
changhua.mch@antgroup.com

**Yinghuan Shi**[†]
Nanjing University
Nanjing, China
syh@nju.edu.cn

## Abstract

In the realm of cross-modal retrieval, seamlessly integrating diverse modalities within multimedia remains a formidable challenge, especially given the complexities introduced by noisy correspondence learning (NCL). Such noise often stems from mismatched data pairs, which is a significant obstacle distinct from traditional noisy labels. This paper introduces Pseudo-Classification based Pseudo-Captioning (PC²) framework to address this challenge. PC² offers a threefold strategy: firstly, it establishes an auxiliary "pseudo-classification" task that interprets captions as categorical labels, steering the model to learn image-text semantic similarity through a non-contrastive mechanism. Secondly, unlike prevailing margin-based techniques, capitalizing on PC²'s pseudo-classification capability, we generate pseudo-captions to provide more informative and tangible supervision for each mismatched pair. Thirdly, the oscillation of pseudo-classification is borrowed to assistant the correction of correspondence. In addition to technical contributions, we develop a realistic NCL dataset called Noise of Web (NoW), which could be a new powerful NCL benchmark where noise exists naturally. Empirical evaluations of PC² showcase marked improvements over existing state-of-the-art robust cross-modal retrieval techniques on both simulated and realistic datasets with various NCL settings. The contributed dataset and source code are released at https://github.com/alipay/PC2-NoiseofWeb.

## CCS Concepts

• **Information systems** → **Multimedia and multimodal retrieval**; • **Computing methodologies** → **Machine learning**.

---

[*]This work was done during the internship at Tiansuan Lab, Ant Group.
[†]Corresponding author.

*MM '24, October 28–November 1, 2024, Melbourne, VIC, Australia.*
© 2024 Copyright held by the owner/author(s). Publication rights licensed to ACM.
ACM ISBN 979-8-4007-0686-8/24/10
https://doi.org/10.1145/3664647.3680860

## Keywords

realistic dataset contribution, image-text retrieval, noisy correspondence learning

**ACM Reference Format:**
Yue Duan, Zhangxuan Gu, Zhenzhe Ying, Lei Qi, Changhua Meng, and Yinghuan Shi. 2024. PC²: Pseudo-Classification Based Pseudo-Captioning for Noisy Correspondence Learning in Cross-Modal Retrieval. In *Proceedings of the 32nd ACM International Conference on Multimedia (MM '24), October 28–November 1, 2024, Melbourne, VIC, Australia.* ACM, New York, NY, USA, 10 pages. https://doi.org/10.1145/3664647.3680860

## 1 Introduction

Cross-modal retrieval, a cornerstone of multimodal learning, is a vibrant domain tasked with bridging diverse modalities in the vast realm of multimedia [36, 52]. Yet, the tangible success of these methods hinges on a critical presumption: the training data must be in harmonious alignment across modalities. The hitch, however, lies in obtaining such perfectly matched data pairs. Manual annotation is not only a huge task but also prone to subjective errors. A potential alternative, often adopted, is mining co-occurring image-text pairs from the vast expanse of the internet [26, 38, 45]. But this convenience comes at a cost: the introduction of noise in the form of mismatched data pairs. This brings us to the crux of our discourse – *noisy correspondence* [25]. Unlike traditional noisy labels, which are about incorrect category labels [32, 35, 48], noisy correspondence is the mismatch between different modalities in paired data (an example is shown in the upper part of Fig. 1). The collected data, riddled with a mix of clean and noisy data pairs, can diminish the effectiveness of cross-modal retrieval techniques [25, 37, 57].

*Noisy correspondence learning* (NCL) mentioned above still holds vast potential for development. Since it is first introduced by NCR [25], only a handful of works have ventured further exploration and they are mainly evaluated on artificially simulated NCL datasets [25, 57]. Thus, we collect 100K website image-meta description pairs from the web to construct a large-scale NCL-specific dataset: **N**oise **of W**eb **(NoW)**, which has more complex, natural, and challenging noisy correspondences. Back to the main topic, the previous NCL solutions can be summarized as adjusting the correspondence labels, which can be recast as the soft margin of triplet loss,

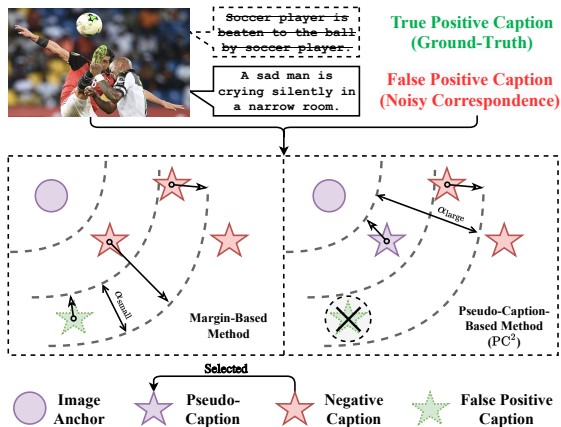

**Figure 1: Illustrations for noisy correspondence and difference between currently popular margin-based methods and proposed pseudo-caption based PC$^2$. PC$^2$ aims to provide direct supervision for false positive pairs with pseudo-caption and larger margin (*i.e.*, $\alpha_{\text{large}}$) in triplet loss, rather than adjusting a smaller margin (*i.e.*, $\alpha_{\text{small}}$) to alleviate the negative influence of false positive pairs in margin-based methods.**

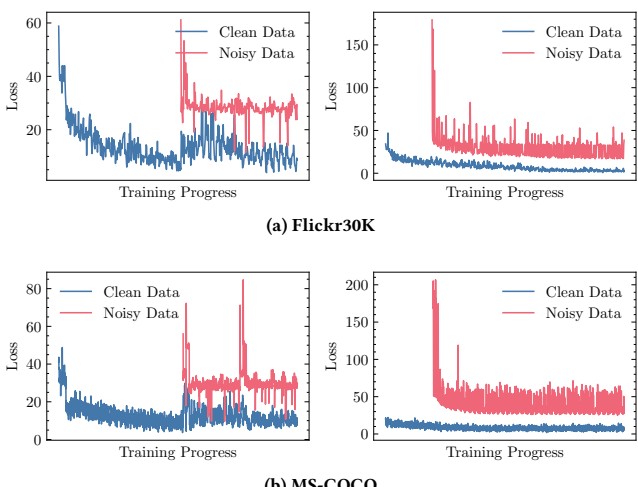

**Figure 2: Experimental results of NCR (left) vs. PC$^2$ (right). Our method shows a more robust learning performance on clean data, maintaining a gradually converging trend with minimal influence from noisy data. In contrast, NCR exhibits a more oscillating pattern in learning clean data, especially when starting to learn from noisy data, causing noticeable fluctuations in the loss of clean data.**

thereby mitigating the negative impact on the training caused by the mismatched image-text pairs [25, 57]. We refer to these methods as margin-based methods, which is showcased on the left side of Fig. 1. Although these methods demonstrates viability and efficacy, it possesses certain limitations. *Adjusting the margin doesn't directly provide beneficial supervisory information for those false positive pairs but rather alleviates their incorrect supervision.*

Meanwhile, although these methods strive to resist noisy data, they are still affected by noticeable adverse impacts, as exemplified in Fig. 2 provided. The learning process of NCR [25] mentioned above presents an oscillatory pattern, especially pronounce during initial encounters with noisy data, thereby inducing notable volatility in the loss associated with clean data. In response to this observed phenomenon, we proffer a novel NCL framework, named **P**seudo-**C**lassification based **P**seudo-**C**aptioning (PC$^2$), engineered for robust cross-modal retrieval with noisy correspondence. This framework can be divided into three integrated solutions:

(1) Inspired by non-contrastive learning [6, 7, 61], we initially design an auxiliary task named "*pseudo-classification*" to reinforce the model's learning from clean data. Simplistically, the caption is interpreted as a categorical label, thereby driving the model to internalize image semantic categories through a refined cross-entropy paradigm. Utilizing cross-entropy loss brings the benefit of a explicit optimization objective for the model, without the need for negative samples. Pseudo-classification enables automatic grouping of visual concepts from image-text pairs, ultimately inducing additional semantic information. (2) Sparked by pseudo-labeling used in various semi-supervised learning tasks [9, 14, 15, 17, 56, 62, 63] and image captioning used in multimodal learning [39, 58], in contrast to the margin-based NCL methods, we propose that offering more informative supervision for each mismatched pair by generating pseudo-captions, which is illustrated in the right side of

Fig. 1. Many studies have highlighted the importance of accurate captions [5, 43, 46]. When mismatched pairs persist in training, their adverse impact on model performance is profound. Consequently, we strive to produce captions for these mismatched pairs as correctly as possible. For specific, capitalizing on the pseudo-classification prowess of PC$^2$, our strategy concurrently compute pseudo-predictions for both clean data and noisy data. These predictions, informed by their intrinsic similarities, serve as the basis for assigning pseudo-captions to the noisy data. Our primary goal remains to ensure correct correspondences across all pairs, thereby steering the model towards a better learning trajectory. (3) We make use of the pseudo-classifier, capitalizing on its oscillatory prediction behavior across different epochs, to perform a simple yet effective correspondence correction for clean data.

In summary, our contributions are as follows: (1) We introduce a NCL-robust framework: PC$^2$, offering a threefold strategy: an auxiliary "pseudo-classification" task using cross-entropy loss; a novel use of pseudo-captions for richer supervision of mismatched pairs, and a correspondence correction mechanism, all rooted in pseudo-classification. (2) PC$^2$ shows promising NCL performance on both the simulated and the realistic datasets, outperforming both popular cross-modal retrieval approaches and NCL-robust methods across a variety of NCL settings. (3) We introduce a realistic dataset Noise of Web (NoW), which can serve as a powerful benchmark for future evaluations of NCL.

## 2 Related Work

Bridging the semantic divide between diverse modalities is the cornerstone in multimedia research [24, 28, 53]. Such cross-modal

endeavors predominantly revolve around mapping these disparate modalities into a unified, learnable space, ensuring measurable semantic correlations. However, the methodologies and challenges associate with this goal vary based on the data modalities and the alignment strategies in play, *e.g.*, image captioning [39, 58], video captioning [51]. For the focus of this article, image-text matching, the crux lies in deriving representations from images and aligning these with their textual counterparts [10, 16, 41].

Although previous image-text matching work has achieved considerable success [8, 12, 33], a recurrent concern in these studies is the assumption of perfectly aligned training data pairs, which is hard to guarantee due to extensive collection and annotation expenses. *Noisy correspondence learning* (NCL), a relatively novel problem, delves into this issue [21, 25, 37, 57]. It addresses the mismatched pairs inaccurately considered positive. Initial research in this domain is NCR [25], which trains image-text matching models robustly with adaptively rectified soft correspondence label. Subsequent to NCR, its successors have ushered in enhancements on NCL. For instance, BiCro [57] introduces an innovative approach to rectify noisy correspondence labels by leveraging the bidirectional cross-modal similarity consistency. This methodology capitalizes on the inherent consistency present within paired data. On the other hand, DECL [37] exploits cross-modal evidential learning to estimate the uncertainty brought by noise to isolate the noisy pairs, A salient feature uniting these methodologies is their conciliatory strategy towards handling misaligned image-text pairs; their designs primarily revolve around mitigating the detrimental impacts of mismatched pairs by isolating them or adjusting a smaller margin in triplet ranking loss. Contrasting these approaches, PC² furnishes direct supervisory signals for images in mismatched pairs, which enriches the learning process.

## 3 Dataset Contribution: Noise of Web

### 3.1 Motivation

The aim of noisy correspondence learning (NCL) is building robust models based on large-scale noisy data, which can be easily obtained on website and apps. However, although there exist some noisy correspondence learning datasets such as MS-COCO [34] and Flickr30K [59] as the benchmarks, the noise in them is human generated and picked, which limits noisy correspondence models' generalization ability towards real-world applications. Randomly replacing some images' caption with others in one dataset is not a perfect choice for noise generating since there may be multiple positive and reasonable captions to one image. Another disadvantage of existing datasets is the huge human labor for writing meaningful captions for images with various different representations. For example, MS-COCO has 616,435 captions for 123,287 images, and all these captions are given by human. Although Conceptual Captions [45] (a realistic datasets) is used for NCL [25], but its low noise ratio (3% ∼ 20%) makes it insufficient for a comprehensive evaluation.

Motivated by the above mentioned, we develop a new dataset named **N**oise **o**f **W**eb (**NoW**) for NCL. It contains 100K cross-modal pairs consisting of website images and multilingual website meta-descriptions (98,000 pairs for training, 1,000 for validation, and 1,000 for testing). NoW has two main characteristics: without human annotations and the noisy pairs are naturally captured.

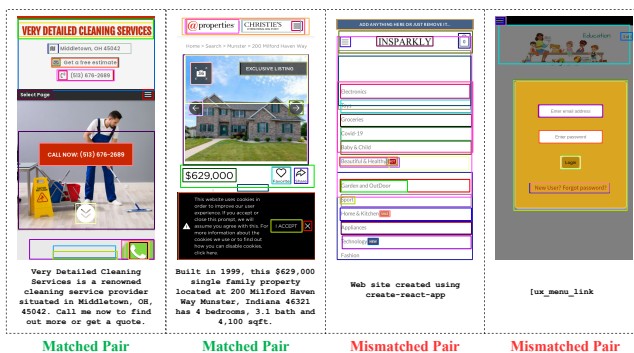

**Figure 3: Sample data pairs in NoW composed of website pages and their corresponding site meta-descriptions. Boxes with different colors are used to display the region proposals obtained by the detection model APT [19] trained by us.**

### 3.2 Data Collection

The source image data of NoW is obtained by taking screenshots when accessing web pages on mobile user interface (MUI) with 720×1280 resolution, and we parse the meta-description field in the HTML source code as the captions. In NCR [25] (predecessor of NCL), each image in all datasets are preprocessed using Faster-RCNN [40] detector provided by [1] to generate 36 region proposals, and each proposal is encoded as a 2048-dimensional feature. Thus, following NCR, we release our the features instead of raw images for fair comparison. However, we can not just use detection methods like Faster-RCNN [40] to extract image features since it is trained on real-world animals and objects on MS-COCO. To tackle this, we adapt APT [19] as the detection model since it is trained on MUI data. Then, we capture the 768-dimensional features of top 36 objects for one image. Using local objects' feature could contribute more to the contrastive learning and pseudo-caption generating, as explained in [12, 25, 31]. Due to the automated and non-human curated data collection process, the noise in NoW is highly authentic and intrinsic. For example, semantic inconsistencies between page content and descriptions (*e.g.*, the third column in Fig. 3), nonsensical garbled description resulting from improper website maintenance (*e.g.*, the fourth column in Fig. 3). The estimated noise ratio of this dataset is nearly **70%**. More details of NoW can be found in Sec. A of Supplementary Material.

## 4 Method

### 4.1 Overview

In the domain of cross-modal retrieval, ensuring accurate correspondence between different modalities, such as images and text, is crucial. To comprehensively study this challenge, we take image-text retrieval as a representative task to delve into the issue of noisy correspondence. At the heart of this task is a training set denoted as $\mathcal{D} = \{(I_i, T_i, c_i)\}_{i=1}^{N}$, where each tuple represents an image-text pair. Here, $I_i$ and $T_i$ are the image and text components of the $i$-th pair, respectively. The label $c_i \in \{0, 1\}$ signifies whether the pair is matched ($c_i = 1$) or mismatched ($c_i = 0$). $N$ represents the total count of data pairs in the training set. In the conventional setting

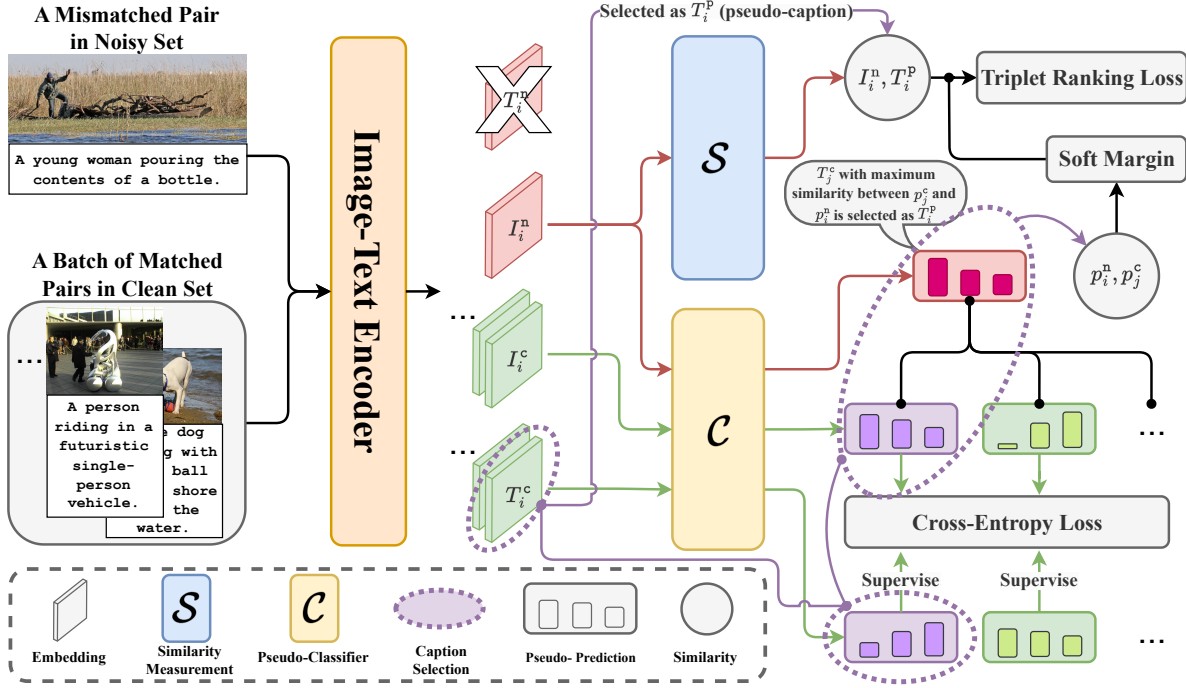

**Figure 4: Visualization of the procedures of *pseudo-classification* and *pseudo-captioning* in PC$^2$. Pseudo-classification: Given a batch of clean data $(I_i^c, T_i^c)$, we first calculate the embeddings of $I_i^c$ and $T_i^c$. Then we use $C$ to obtain their pseudo-predictions $p_i^c$ and $q_i^c$, respectively. $q_i^c$ is used as the classification label to supervise the training of $C$ on $p_i^c$ using the standard cross-entropy loss function, in hopes of reinforcing the training of image-text matching. Pseudo-captioning: Given noisy data $(I_i^n, T_i^n)$, we first discard its caption $T_i^n$. We input the embedding of $I_i^n$ into $C$ to obtain its pseudo-prediction $p_i^n$, then find the most similar one (denoted as $p_j^c$) to $p_i^n$ among $p_i^c$ from the aforementioned batch of clean data being trained synchronously. We assign the corresponding caption of $p_j^c$ (i.e., $T_j^c$) to $I_i^n$ as the pseudo-caption $T_i^p$, and also utilize a margin based on pseudo-prediction similarity to train the matching model with a triplet ranking loss.**

of image-text retrieval, it is often assumed that all image-text pairs in the dataset are matching (*i.e.*, $\forall i \in \{1, \cdots, N\}, c_i = 1$). However, multimodal datasets might be imprecisely annotated in real-world, especially if they are sourced from the internet or created using cost-effective methods (*i.e.*, $\exists i \in \{1, \cdots, N\}, c_i = 0$), which we refer to as *noisy correspondence learning* (NCL). In general, we do not have sufficient resources to accurately identify the matching status of all image-text pairs, as $c_i$ can be considered inaccessible.

Given $\mathcal{D}$, we use two modal-specific encoder $f(\cdot)$ and $g(\cdot)$ to respectively compute the feature embedding $f(I)$ and $g(T)$. The fundamental aim of cross-modal retrieval is to map different modalities into a unified feature space, where positive pairs should exhibit higher feature similarities, while negative pairs should manifest lower similarities. The similarity between given image-text pairs is determined using the function $S(I, T)$, which is a shorthand for $S(f(I), g(T))$. Generally, the primary objective is to optimize $f$ and $g$ by minimizing a triplet ranking loss function, which is influenced by the similarity measure and a distance margin $\alpha$:

$$\mathcal{L}^t(I_i, T_i) = [\alpha - S(I_i, T_i) + S(I_i, \tilde{T}_h)]_+ + [\alpha - S(I_i, T_i) + S(\tilde{I}_h, T_i)]_+,$$
$$(1)$$

where $[x]_+ = \max(x, 0)$, $(I_i, T_i)$, $(I_i, \tilde{T}_h)$ and $(\tilde{I}_h, T_i)$ are the positive pair, negative pair treating image as query and negative pair treating text as query, respectively. $\tilde{I}_h = \arg\max_{I_j \neq I_i} S(I_j, T_i)$ and $\tilde{T}_h = \arg\max_{T_j \neq T_i} S(I_i, T_j)$ are the hardest negatives in the minibatch [16]. Dynamic margin plays a crucial role in NCL. Previous margin-based approaches [25, 37, 57] mitigate the impact of mismatched pairs on model training by cleverly adjusting it. The general adjustment strategy is to set a larger $\alpha$ for matched pairs and a smaller $\alpha$ for mismatched pairs. However, our focus is on $T_i$, *i.e.*, we aim to ensure that all images in the pairs have the correct corresponding captions., as optimizing this loss will help the model converge towards a better direction.

In NCL, both noisy and clean data are intermixed. Therefore, the first thing we need to consider is how to distinguish the two as correctly as possible. For simplicity, we directly utilize the memorization effect[1] based *co-dividing* module in [25] to predict the clean probability $w_i$ of $(I_i, T_i, c_i) \in \mathcal{D}$. Setting a threshold $\tau$, we divide $\mathcal{D}$ into clean subset $\mathcal{D}^c = \{(I_i^c, T_i^c, c_i)\}_{i=1}^{N^c}$ and noisy subset $\mathcal{D}^n = \{(I_i^n, T_i^n, c_i)\}_{i=1}^{N^n}$, *i.e.*, $\mathcal{D} = \mathcal{D}^c \cup \mathcal{D}^n$ and $\mathcal{D}^c \cap \mathcal{D}^n = \emptyset$. For

---

[1]Deep neural networks (DNNs) tend to have relatively high loss for the noisy data and relatively low loss for the noisy clean in the training [2].

specific, $(I_i, T_i, c_i)$ with $w_i > \tau$ is selected into $\mathcal{D}^c$, otherwise it is selected into $\mathcal{D}^n$. In this paper, we refer to $(I_i^c, T_i^c) \in \mathcal{D}^c$ as *clean data*, while $(I_i^n, T_i^n) \in \mathcal{D}^n$ is referred to as *noisy data* (because $c_i$ is solely used for defining the task of NCL, it will be omitted in the subsequent discussions). Furthermore, following [25, 57], we adopt the *co-training* manner [20] to alleviate the error accumulation problem. Due to space limitation, the details of *co-dividing* and *co-training* are placed in Sec. B of Supplementary Material.

## 4.2 Pseudo-Classification

In NCL, addressing mismatched data is paramount. However, many approaches often overlook the protection of learning from clean data. As previously discussed in Sec. 1, once mismatched pairs are introduced into training, the efforts invested in learning from clean data can be significantly compromised. To enhance the robustness of training on clean data, we propose an auxiliary training task that reinforces the learning of such data. A key insight we offer is that in image-text pairs, the caption of an image can be considered as a classification label $y \in \{1, \cdots, K\}$, where $K$ is a pre-defined hyper-parameter. Hence, training on image-text pairs can be conceptualized as an $K$-way classification task. For instance, we can categorize the captions in the dataset into two main classes (*i.e.*, $K = 2$): descriptions of natural landscapes and descriptions of biological actions. We aim to train the model to group images of natural landscapes and images containing living organisms into their respective classes. To achieve this goal, we set up a *pseudo-classifier* $C(\cdot)$ and utilize the captions in clean data to generate pseudo-labels for the training of $C$.

Specifically, given a mini-batch of clean data $\{(I_i^c, T_i^c)\}_{i=1}^B$ with batch size $B$, we firstly compute pseudo-predictions $p_i^c = C(f(I_i^c))$ and $q_i^c = C(g(T_i^c))$, where $p_i^c, q_i^c \in \mathbb{R}_+^K$ are probability vectors (*i.e.*, soft label). Next, we conduct cross-entropy loss between the hard pseudo-labels $\hat{q}_i^c = \arg\max(q_i^c)$ and the pseudo-predictions of images (*i.e.*, $p_i^c$):

$$\mathcal{L}^{\text{pse}} = \frac{1}{B} \sum_{i=1}^B H(\hat{q}_i^c, p_i^c), \tag{2}$$

where $H(P, Q)$ denotes the standard cross-entropy loss between distribution $Q$ and $P$. The hard pseudo-label is widely leveraged in semi-supervised learning [13, 47] to achieve entropy minimization [18], which encourages the model to make highly confident predictions. Moreover, to avoid $C$ from assigning all samples to a single class, we minimize an entropy loss to spreads the pseudo-predictions uniformly across the all classes [3, 4, 50]:

$$\mathcal{L}^{\text{ent}} = -\frac{1}{B} \sum_{i=1}^B p_i^c \log(\frac{1}{B} \sum_{i=1}^B p_i^c). \tag{3}$$

Our pseudo-classification loss additionally helps the model capture similarity relationships between samples. It strengthens the model's learning from clean data in $\mathcal{D}^c$, enhancing its ability to resist the interference of noisy data in $\mathcal{D}^n$.

## 4.3 Pseudo-Prediction Based Pseudo-Captioning

The framework of PC$^2$ is shown in Fig. 4. With pseudo-classifier $C$, we design a simple and effective approach to assign pseudo-captions to $I_i^n$. Given a mini-batch of data $\{(I_i^c, T_i^c), (I_i^n, T_i^n)\}_{i=1}^B$, we first compute their pseudo-predictions $p_i^c = C(f(I_i^c))$ and $p_i^n = C(f(I_i^n))$ for $I_i^c$ and $I_i^n$. Then, for each $I_i^n$, we assign the pseudo-caption $T_i^p$ by

$$T_i^p = T_j^c \quad \text{with} \quad j = \underset{b \in \{1, \cdots, B\}}{\arg\max} (S^p(p_i^n, p_b^c)), \tag{4}$$

where $S^p(\cdot, \cdot)$ is a function that can be used to compute the similarity between two distributions. Then, we assemble $I_i^n$ and $T_i^p$ into a pseudo-pair $(I_i^n, T_i^p)$ and substitute them into Eq. (1), aiming to provide more accurate supervision signals for model training. As we cannot guarantee that the found pseudo-caption accurately reflects the semantic information of $I_i^n$, we dynamically adjust the margin to ensure that the model benefits from a more accurate level of correspondence during training. For specific, we adaptively adjust $\alpha$ in Eq. (1) with selected $j$ in Eq. (4) :

$$\alpha^n = \frac{m^{S^p(p_i^n, p_j^c)} - 1}{m - 1} \alpha, \tag{5}$$

where $m$ is a pre-defined curve parameter. The underlying principle here is that if the similarity of the pseudo-predictions $S^p(p_i^n, p_j^c)$ is higher, then the similarity between $I_i^n$ and $I_j^c$ (*i.e.*, the image in the original pair where $T_i^p$ is present) should also be higher, indicating a stronger correspondence between $I_i^n$ and $T_i^p$.

Then, for the noisy data $\{(I_i^n, T_i^n)\}_{i=1}^B$ in the given mini-batch, we train the model by minimizing the following loss:

$$\mathcal{L}^n = \sum_{i=1}^B \left( [\alpha^n - S(I_i^n, T_i^p) + S(I_i^n, \tilde{T}_h^p)]_+ \right.$$
$$\left. + [\alpha^n - S(I_i^n, T_i^p) + S(\tilde{I}_h^n, T_i^p)]_+ \right). \tag{6}$$

## 4.4 Prediction Oscillation Based Correspondence Rectification

In addition to paying special attention to noisy data, learning from clean data cannot be taken lightly, because we cannot guarantee that mismatched pairs have not been erroneously included in $\mathcal{D}^c$. Thus, we introduce a correspondence correction module with the following core idea: the pseudo-classification results of images, learned from pseudo-labels based on captions with correct correspondences, should be stable, *i.e.*, *oscillating pseudo-predictions indicate low correspondence in the image-caption pair*.

We define *prediction oscillation* as the difference between predictions for the same sample between adjacent epochs. A larger difference indicates a higher oscillation, indicating that the model is less confident about the sample and is resisting the supervision provided by the caption-based classification labels, *i.e.*, implying a weaker correspondence between the image and caption. This pattern is very similar to the DNN's memorization effect mentioned in Sec. 4.1. Let $p_i^{c,(e)}$ represent the pseudo-prediction at epoch $e$, and its prediction oscillation $o_i^{(e)}$ is evaluated by:

$$o_i^{(e)} = D_{KL}(p_i^{c,(e-1)} \| p_i^{c,(e)}), \tag{7}$$

where $D_{KL}(P \parallel Q)$ is the KL-divergence between distribution $Q$ and $P$. We input $\{o_i^{(e)}\}_{i=1}^B$ into the co-dividing module described in Sec. 4.1 and obtain the prediction oscillation based clean probabilities $\{w_i^o\}_{i=1}^B$. Following Eq. (5), we recast the strength of correspondence to the margin of Eq. (1), to assist the learning of clean data, *i.e.*,

$$\alpha^c = \frac{m^{w_i^c + (1 - w_i^c)\mathbb{1}(w_i^o \geq \tau)w_i^o} - 1}{m - 1}\alpha, \tag{8}$$

where $\mathbb{1}(\cdot)$ is the indicator function. More explanations of Eqs. (7) and (8) can be found in Sec. B of Supplementary Material. Next, for the clean data $\{(I_i^c, T_i^c)\}_{i=1}^B$ in the given mini-batch, we minimize the following loss:

$$\mathcal{L}^c = \sum_{i=1}^B \left( [\alpha^c - S(I_i^c, T_i^c) + S(I_i^c, \tilde{T}_h^c)]_+ \right.$$
$$\left. + [\alpha^c - S(I_i^c, T_i^c) + S(\tilde{I}_h^c, T_i^c)]_+ \right). \tag{9}$$

In sum, the total loss of PC$^2$ can be presented as

$$\mathcal{L} = \mathcal{L}^c + \lambda^n \mathcal{L}^n + \lambda^{pse} \mathcal{L}^{pse} + \lambda^{ent} \mathcal{L}^{ent}, \tag{10}$$

where $\lambda^n$, $\lambda^{pse}$ and $\lambda^{ent}$ are pre-defined loss weights.

## 5 Experiment

### 5.1 Experimental Setup

**Datasets.** We mainly conduct experiments on two prominent image-text retrieval datasets and our proposed realistic NCL benchmark: (1) Flickr30K [59]: This dataset encompasses 31,000 images, each coupled with five captions. The data is partitioned into 29,000 image-text pairs for training, 1,000 for validation, and 1,000 for testing. (2) MS-COCO [34]: Consisting of 123,287 images, each image in this dataset is accompanied by five captions. The division is as follows: 113,287 image-text pairs for training, 5,000 for validation, and 5,000 for testing. (3) Noise of Web: Please refer to Sec. 3 for details. Moreover, the additional result on realistic dataset Conceptual Captions [45] can be found in Sec. C.1 of Supplementary Material.

**Performance Metrics.** The primary metric for assessing retrieval performance is the recall rate at $k$ (R@$k$). We use both images and text as query entities and report on R@1, R@5, and R@10 for the evaluation. For the well-annotated datasets Flickr30K and MS-COCO, we introduce artificial noise by randomly mixing the training images and captions at five noise levels: 0%, 20%, 40%, 50%, and 60%. For all evaluations, the best checkpoint is selected based on the validation set, and its test set performance is reported.

**Baselines.** For a comprehensive comparison, we extensively employ the following baselines: (1) generic image-text matching approaches: SCAN [31], VSRN [33], IMRAM [8], SASGR, SGRAF [12] (specially, SGR$^*$ and SGR-C [25] are SGR pre-training without hard negatives and SGR training on clean data without noisy data, respectively) and (2) noisy-correspondence-resistant techniques: NCR [25], DECL [37], BiCro [57] and L2RM [22].

### 5.2 Implementation Details

Just like the previous state-of-the-art (SOTA) NCL methods [25, 57], PC$^2$ can also be universally extended to various cross-modal

retrieval models. For a fair comparison, we adopt the same cross-modal retrieval backbone, SGR [12], as used in [25, 57], *i.e.*, a full-connected layer is adopted for $f(\cdot)$, Bi-GRU [42] is adopted for $g(\cdot)$ and a graph reasoning technique proposed in [30] is adopted for $S(\cdot, \cdot)$. Similarly, the training details (*e.g.*, batch size $B = 128$, threshold $\tau = 0.5$, margin $\alpha = 0.2$, $m = 10$) are kept consistent with [25, 57]. For the additional hyper-parameters in PC$^2$, we set $K = 128$ for pseudo-classification and adopt cosine similarity for $S^p$ used in pseudo-captioning. For loss weight, we set $\lambda^n = \lambda^{pse} = 1$ and $\lambda^{ent} = 10$. Following [25], we firstly warm up the model for 5, 10 and 10 epochs for Flickr30K, MS-COCO and NoW, respectively. Then, we train the model for 50 epochs in all experiments. We use the same Adam optimizer [27] with the default parameters for training as in [25, 57]. The complete list of hyper-parameters can be found in Sec. B of Supplementary Material.

### 5.3 Results and Analysis

**Main Results.** We summarize the main comparisons in Tab. 1, where SoC shows promising results on both Flickr30K and MS-COCO. In the most NCL settings, PC$^2$ outperforms all baseline methods on the indicator Rsum by a tangible margin, *e.g.*, PC$^2$ outperforms the best baseline method on Flickr30K at noise ratios of 40%, 50%, and 60% by 3.3, 10.2 and 5.9, respectively.

Further, it is noteworthy that even in settings without noisy correspondences, PC$^2$ still achieves competitive performance, which to some extent outperform the best generic method: SGRAF (504.8 vs. 499.6 on Flickr30K). Conversely, NCR may be defeated by SGRAF (522.5 vs. 524.3 on MS-COCO). From the perspective of general image-text matching methods, they all suffer a significant setback at high noise ratios (*e.g.*, NCR with 60%), highlighting the importance of NCL methods. From the viewpoint of noise-robust methods, margin-based approaches are generally weaker than the pseudo-caption-based PC$^2$. The core enhancement of our method lies in its ability to provide the correct supervisory signal for mismatched pairs as much as possible, enabling the model to make better use of noisy data. This offers a richer imagination space for NCL. Although the pseudo-captions assigned by PC$^2$ may not be completely consistent with the semantics of the noisy images, a certain degree of semantic overlap is sufficient to provide effective supervision. Additionally, we show the comparison with BiCro$^*$ [57], a variant of BiCro that uses mismatch thresholds to filter out mismatched pairs (the performance of PC$^2$ can also benefit from this technique), in Sec. C.2 of Supplementary Material.

**Results on NoW.** Results on our challenging NCL benchmark, NoW, in Tab. 2, show our method's consistent performance advantage. Since a significant portion of captions in NoW are in Chinese, we first consider using JiebaTokenizer [23] to conduct tokenization. Moreover, we provide the additional results on BPETokenizer [44] and BertTokenizer [11, 55] that can be applied to multilingual texts in Sec. C.3 of Supplementary Material. Compared to meticulously organized datasets like MS-COCO, NoW better mirrors real-world industry scenarios. The lower success of existing methods on NoW reveals NCL research gaps, opening new exploration avenues for the community. Challenges of NoW are twofold: **(1)** high noise levels and sparse visual elements in images (web pages), with overly verbose or less informative captions; **(2)** overly abstract

**Table 1: Performance comparison of image-text retrieval on Flickr30K and MS-COCO with recall at 1, 5, and 10 (R@1, R@5, R@10), along with Rsum (sum of recall values). We mark out the best results in bold and the second best results in underline. For a fair comparison, we adopt the noise ratio protocol of NCR (0%, 20% and 50%) [25] and BiCro (20%, 40% and 60%) [57].**

| Noise | Methods | Flickr30K | | | | | | | MS-COCO | | | | | | |
| | | Image → Text | | | Text → Image | | | | Image → Text | | | Text → Image | | | |
| | | R@1 | R@5 | R@10 | R@1 | R@5 | R@10 | Rsum | R@1 | R@5 | R@10 | R@1 | R@5 | R@10 | Rsum |
|---|---|---|---|---|---|---|---|---|---|---|---|---|---|---|---|
| 0% | SCAN [31] | 67.4 | 90.3 | 95.8 | 48.6 | 77.7 | 85.2 | 465.0 | 69.2 | 93.6 | 97.6 | 56.0 | 86.5 | 93.5 | 496.4 |
| | VSRN [33] | 71.3 | 90.6 | 96.0 | 54.7 | 81.8 | 88.2 | 482.6 | 76.2 | 94.8 | 98.2 | 62.8 | 89.7 | 95.1 | 516.8 |
| | IMRAM [8] | 74.1 | 93.0 | 96.6 | 53.9 | 79.4 | 87.2 | 484.2 | 76.7 | 95.6 | 98.5 | 61.7 | 89.1 | 95.0 | 516.6 |
| | SAF [12] | 73.7 | 93.3 | 96.3 | 56.1 | 81.5 | 88.0 | 488.9 | 76.1 | 95.4 | 98.3 | 61.8 | 89.4 | 95.3 | 516.3 |
| | SGR [12] | 75.2 | 93.3 | 96.6 | 56.2 | 81.0 | 86.5 | 488.8 | 78.0 | 95.8 | 98.2 | 61.4 | 89.3 | 95.4 | 518.1 |
| | SGRAF [12] | 77.8 | 94.1 | 97.4 | 58.5 | 83.0 | 88.8 | 499.6 | **79.6** | 96.2 | 98.5 | 63.2 | 90.7 | **96.1** | **524.3** |
| | NCR [25] | 77.3 | 94.0 | **97.5** | 59.6 | **84.4** | **89.9** | 502.7 | 78.7 | 95.8 | 98.5 | 63.3 | 90.4 | 95.8 | 522.5 |
| | PC² (Ours) | **78.7** | **94.8** | 97.0 | **60.0** | **84.4** | 89.8 | **504.8** | 79.1 | **96.5** | **98.8** | **64.0** | 90.3 | 95.6 | **524.3** |
| 20% | SCAN [31] | 59.1 | 83.4 | 90.4 | 36.6 | 67.0 | 77.5 | 414.0 | 66.2 | 91.0 | 96.4 | 45.0 | 80.2 | 89.3 | 468.1 |
| | VSRN [33] | 58.1 | 82.6 | 89.3 | 40.7 | 68.7 | 78.2 | 417.6 | 25.1 | 59.0 | 74.8 | 17.6 | 49.0 | 64.1 | 289.6 |
| | IMRAM [8] | 63.0 | 86.0 | 91.3 | 41.4 | 71.2 | 80.5 | 433.4 | 68.6 | 92.8 | 97.6 | 55.7 | 85.0 | 91.0 | 490.7 |
| | SAF [12] | 51.0 | 79.3 | 88.0 | 38.3 | 66.5 | 76.2 | 399.3 | 67.3 | 92.5 | 96.6 | 53.4 | 84.5 | 92.4 | 486.7 |
| | SGR* [12] | 62.8 | 86.2 | 92.2 | 44.4 | 72.3 | 80.4 | 438.3 | 67.8 | 91.7 | 96.2 | 52.9 | 83.5 | 90.1 | 482.2 |
| | SGR-C [12] | 72.8 | 90.8 | 95.4 | 56.4 | 82.1 | 88.6 | 486.1 | 75.4 | 95.2 | 97.9 | 60.1 | 88.5 | 94.8 | 511.9 |
| | NCR [25] | 75.0 | 93.9 | 97.5 | 58.3 | 83.0 | 89.0 | 496.7 | 77.7 | 95.5 | 98.2 | 62.5 | 89.3 | 95.3 | 518.5 |
| | DECL [37] | 75.4 | 93.2 | 96.2 | 56.8 | 81.7 | 88.4 | 491.7 | 76.9 | 95.3 | 98.2 | 61.3 | 89.0 | 95.1 | 515.8 |
| | BiCro [57] | 78.3 | 94.1 | 97.3 | **60.0** | 83.7 | 89.5 | 502.9 | 78.2 | 95.9 | 98.4 | 62.5 | 89.8 | **95.5** | 520.3 |
| | L2RM [22] | 77.9 | **95.2** | **97.8** | 59.8 | 83.6 | 89.5 | **503.8** | **80.2** | **96.3** | **98.5** | **64.2** | **90.1** | 95.4 | **524.7** |
| | PC² (Ours) | **78.7** | 94.9 | 96.9 | 59.8 | **83.9** | **89.6** | **503.8** | 77.8 | 95.7 | 98.4 | 62.8 | 89.7 | 95.3 | 519.7 |
| 40% | SCAN [31] | 26.0 | 57.4 | 71.8 | 17.8 | 40.5 | 51.4 | 264.9 | 42.9 | 74.6 | 85.1 | 24.2 | 52.6 | 63.8 | 343.2 |
| | VSRN [33] | 2.6 | 10.3 | 14.8 | 3.0 | 9.3 | 15.0 | 55.0 | 29.8 | 62.1 | 76.6 | 17.1 | 46.1 | 60.3 | 292.0 |
| | IMRAM [8] | 5.3 | 25.4 | 37.6 | 5.0 | 13.5 | 19.6 | 106.4 | 51.8 | 82.4 | 90.9 | 38.4 | 70.3 | 78.9 | 412.7 |
| | SAF [12] | 7.4 | 19.6 | 26.7 | 4.4 | 12.2 | 17.0 | 87.3 | 13.5 | 43.8 | 48.2 | 16.0 | 39.0 | 50.8 | 211.3 |
| | SGR [12] | 4.1 | 16.6 | 24.1 | 4.1 | 13.2 | 19.7 | 81.8 | 10.3 | 38.4 | 50.2 | 11.4 | 34.5 | 41.5 | 186.3 |
| | SGRAF [12] | 8.3 | 18.1 | 31.4 | 5.3 | 16.7 | 21.3 | 101.1 | 15.8 | 23.4 | 54.6 | 17.8 | 43.6 | 54.1 | 209.3 |
| | NCR [25] | 68.1 | 89.6 | 94.8 | 51.4 | 78.4 | 84.8 | 467.1 | 74.7 | 94.6 | 98.0 | 59.6 | 88.1 | 94.7 | 509.7 |
| | DECL [37] | 69.0 | 90.2 | 94.8 | 50.7 | 76.3 | 84.1 | 465.1 | 73.6 | 94.6 | 97.9 | 57.8 | 86.9 | 93.9 | 504.7 |
| | BiCro [57] | 73.6 | 93.0 | 96.4 | 56.0 | 80.8 | 87.4 | 487.2 | 76.4 | 95.2 | **98.6** | 61.5 | **89.4** | **95.5** | 516.6 |
| | L2RM [22] | **75.8** | 93.2 | **96.9** | 56.3 | 81.0 | 87.3 | 490.5 | **77.5** | 95.8 | 98.4 | 62.0 | 89.1 | 94.9 | 517.7 |
| | PC² (Ours) | **75.8** | 93.5 | **96.9** | **57.5** | 81.9 | 88.2 | **493.8** | 77.4 | 95.8 | 98.4 | 62.1 | 89.4 | 95.1 | **518.2** |
| 50% | SCAN [31] | 27.7 | 57.6 | 68.8 | 16.2 | 39.3 | 49.8 | 259.4 | 40.8 | 73.5 | 84.9 | 5.4 | 15.1 | 21.0 | 240.7 |
| | VSRN [33] | 14.3 | 37.6 | 50.0 | 12.1 | 30.0 | 39.4 | 183.4 | 23.5 | 54.7 | 69.3 | 16.0 | 47.8 | 65.9 | 277.2 |
| | IMRAM [8] | 9.1 | 26.6 | 38.2 | 2.7 | 8.4 | 12.7 | 97.7 | 21.3 | 60.2 | 75.9 | 22.3 | 52.8 | 64.3 | 296.8 |
| | SAF [12] | 30.3 | 79.3 | 88.0 | 38.3 | 66.5 | 76.2 | 378.6 | 67.3 | 92.5 | 96.6 | 53.4 | 84.5 | 92.4 | 486.7 |
| | SGR* [12] | 36.9 | 68.1 | 80.2 | 29.3 | 56.2 | 67.0 | 337.7 | 67.0 | 87.4 | 93.6 | 46.0 | 74.2 | 79.0 | 447.2 |
| | SGR-C [12] | 69.8 | 90.3 | 94.8 | 50.1 | 77.5 | 85.2 | 467.7 | 71.7 | 94.1 | 97.7 | 57.0 | 86.6 | 93.7 | 500.8 |
| | NCR [25] | 72.9 | 93.0 | 96.3 | **54.3** | 79.8 | 86.5 | 482.8 | 74.6 | 94.6 | 97.8 | 59.1 | 86.6 | 94.5 | 507.2 |
| | DECL [37] | 71.3 | 90.7 | 94.6 | 52.2 | 78.7 | 86.0 | 473.5 | 74.4 | 94.2 | 98.0 | 58.8 | 87.6 | 94.3 | 507.3 |
| | PC² (Ours) | **74.9** | **91.5** | **95.7** | **54.3** | **80.2** | **87.1** | **483.7** | **76.1** | **95.5** | **98.4** | **60.9** | **88.7** | **94.6** | **514.2** |
| 60% | SCAN [31] | 13.6 | 36.5 | 50.3 | 4.8 | 13.6 | 19.8 | 138.6 | 29.9 | 60.9 | 74.8 | 0.9 | 2.4 | 4.1 | 173.0 |
| | VSRN [33] | 0.8 | 2.5 | 5.3 | 1.2 | 4.2 | 6.9 | 20.9 | 11.6 | 34.0 | 47.5 | 4.6 | 16.4 | 25.9 | 140.0 |
| | IMRAM [8] | 1.5 | 8.9 | 17.4 | 1.9 | 5.0 | 7.8 | 42.5 | 18.2 | 51.6 | 68.0 | 17.9 | 43.6 | 54.6 | 253.9 |
| | SAF [12] | 0.1 | 1.5 | 2.8 | 0.4 | 1.2 | 2.3 | 8.3 | 0.1 | 0.5 | 0.7 | 0.8 | 3.5 | 6.3 | 11.9 |
| | SGR [12] | 1.5 | 6.6 | 9.6 | 0.3 | 2.3 | 4.2 | 24.5 | 0.1 | 0.6 | 1.0 | 0.1 | 0.5 | 1.1 | 3.4 |
| | SGRAF [12] | 2.3 | 5.8 | 10.9 | 1.9 | 6.1 | 8.2 | 35.2 | 0.2 | 3.6 | 7.9 | 1.5 | 5.9 | 12.6 | 31.7 |
| | NCR [25] | 13.9 | 37.7 | 50.5 | 11.0 | 30.1 | 41.4 | 184.6 | 0.1 | 0.3 | 0.4 | 0.1 | 0.5 | 1.0 | 2.4 |
| | DECL [37] | 64.5 | 85.8 | 92.6 | 44.0 | 71.6 | 80.6 | 439.1 | 69.7 | 93.4 | 97.5 | 54.5 | 85.2 | 92.6 | 492.9 |
| | BiCro [57] | 68.3 | 90.4 | 93.8 | 51.9 | 76.9 | 84.4 | 465.7 | 73.9 | **94.7** | 97.7 | 58.7 | 87.0 | **93.8** | 505.8 |
| | L2RM [22] | 70.0 | **90.8** | 95.4 | 51.3 | 76.4 | 83.7 | 467.6 | **75.4** | **94.7** | **97.9** | **59.2** | 87.4 | **93.8** | **508.4** |
| | PC² (Ours) | **70.8** | 90.3 | **94.4** | **53.1** | **79.0** | **85.9** | **473.5** | 74.2 | 94.4 | 97.8 | 58.9 | **87.5** | **93.8** | 506.6 |

**Table 2: Performance comparison of image-text retrieval on NoW with captions tokenized by JiebaTokenizer [55].**

| Methods | Image → Text | | | Text → Image | | | Rsum |
|---|---|---|---|---|---|---|---|
| | R@1 | R@5 | R@10 | R@1 | R@5 | R@10 | |
| SGR [12] | 11.0 | 25.3 | 34.5 | 11.3 | 24.7 | 34.1 | 140.9 |
| NCR [25] | 13.8 | 27.6 | 35.6 | 13.6 | 27.7 | 34.4 | 152.7 |
| DECL [37] | 14.2 | 27.9 | 35.8 | 13.5 | 28.2 | 35.4 | 155.0 |
| BiCro [57] | 14.6 | 28.3 | 36.4 | 13.1 | 28.7 | 36.5 | 157.6 |
| PC$^2$ (Ours) | **16.0** | **29.5** | **36.9** | **15.5** | **29.0** | **37.0** | **163.8** |

**Table 3: Ablation studies on the three components in PC$^2$. The experiments are conducted Flickr30K with 40% noise.**

| Components | | | Image → Text | | | Text → Image | | | Rsum |
|---|---|---|---|---|---|---|---|---|---|
| P-Cls | P-Cap | CR | R@1 | R@5 | R@10 | R@1 | R@5 | R@10 | |
| ✓ | ✓ | ✓ | **75.8** | **93.5** | **96.9** | **57.5** | **81.9** | **88.2** | **493.8** |
| ✓ | ✓ | | 75.0 | 93.1 | 96.0 | 56.9 | 81.2 | 87.6 | 489.8 |
| ✓ | | ✓ | 71.8 | 91.7 | 95.9 | 53.6 | 80.0 | 86.8 | 479.8 |
| ✓ | | | 71.2 | 91.3 | 95.5 | 53.4 | 79.1 | 86.4 | 476.9 |
| | ✓ | | 70.1 | 90.3 | 95.1 | 51.9 | 78.5 | 85.2 | 471.1 |

**Table 4: Ablation studies on $K$ (i.e., the class number of $C$). The experiments are conducted Flickr30K with 40% noise.**

| $K$ | Image → Text | | | Text → Image | | | Rsum |
|---|---|---|---|---|---|---|---|
| | R@1 | R@5 | R@10 | R@1 | R@5 | R@10 | |
| 32 | 72.7 | 90.9 | 93.9 | 55.4 | 80.1 | 86.4 | 479.5 |
| 128 | **75.8** | **93.5** | **96.9** | **57.5** | **81.9** | **88.2** | **493.8** |
| 512 | 75.1 | 91.8 | 94.7 | 58.4 | 80.5 | 87.5 | 488.0 |
| 2048 | 75.4 | 90.7 | 93.5 | 57.8 | 80.7 | 86.2 | 484.3 |
| 8192 | 71.1 | 89.9 | 94.0 | 53.2 | 79.1 | 85.8 | 473.0 |

captions even in correctly matched samples, *e.g.*, the fifth pairs in the first and second row of Fig. 6 in Supplementary Material. The excessive noise in NoW might be merely superfluous for margin-based methods. In contrast, pseudo-caption-based PC$^2$ could better enable the effective utilization of numerous mismatched pairs.

### 5.4 Ablation Study

The construction of PC$^2$ relies on components *pseudo-classification* (P-Cls), *pseudo-captioning* (P-Cap), and *correspondence rectification* (CR). We conduct ablation experiments on these three components to demonstrate their effectiveness. As shown in Tab. 3, the default PC$^2$ (the first row) achieves the most superior performance compared with other settings. The second and third rows illustrate the effectiveness of P-Cap and CR, respectively, while the fourth row indicates that even without transforming margin-based methods into pseudo-caption-based methods, relying solely on P-Cls can still greatly benefit the model from enhanced learning on clean data. Furthermore, to explore the effectiveness of P-Cls based pseudo-captioning, we strip away P-Cls and employ an alternative, more straightforward implementation to assign pseudo-captions (the fifth

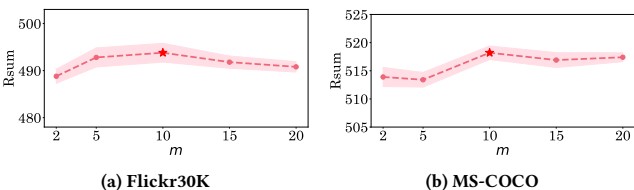

**(a) Flickr30K**        **(b) MS-COCO**

**Figure 5: Ablation studies on curve parameter $m$ on both Flickr30K and MS-COCO with 40% noise.**

row): directly calculating the similarity between the embeddings of noisy and clean data images within the same batch and, like PC$^2$, using the most similar pair to transfer the pseudo-caption. Default PC$^2$ exceeds this design, demonstrating that learning through pseudo-classification can more effectively refine the semantic information of images, thereby aiding in more rational pseudo-captioning.

As P-Cls is the foundation of all PC$^2$'s components, we should carefully select the value of $K$. As shown in Tab. 4, a moderately sized $K$ allows $C$ to learn the images with appropriate granularity, thereby better enhancing the learning from clean data and the selection of pseudo-captions. Following previous methods [21, 25, 57], we set curve parameter $m = 10$ for the margin adjustment functions Eqs. (5) and (8) in PC$^2$. As shown in Fig. 5, we verify the suitability of this default setting for $m$. More ablations on other hyper-parameters can be found in Sec. C.4 of Supplementary Material.

## 6 Discussion and Future Work

**Methodology.** The design of batch-level pseudo-caption search is a trade-off between ease of implementation, efficiency, and performance. A larger batch size could make it easier for PC$^2$ to find the appropriate pseudo-captions, thereby improving the performance. Likewise, global search for pseudo-captions could further enhance PC$^2$. In our future work, improving the caption search space of PC$^2$ or using image captioning solution for noisy data is our focus. In addition, it is an indisputable fact that vision-language model [38, 60] and multimodal large language model [29, 49, 54] have great potential as backbones in cross-modal retrieval tasks, but their application in NCL has not been fully explored [25]. This will also be our future direction of progress.

**Dataset.** In the future, we will increase the overall size of our dataset, and improve the validation and test sets by manually re-annotating the captions of the images, rather than just picking pairs that are manually considered to match from the original dataset.

## 7 Conclusion

In this paper, we introduce **P**seudo-**C**lassification based **P**seudo-**C**aptioning (PC$^2$) framework to enhance cross-modal retrieval in the presence of noisy correspondence learning. PC$^2$ innovatively employs pseudo-classification and pseudo-captions for richer supervision of mismatched pairs and experiments showcases PC$^2$'s superiority over existing techniques. This study further contributes by open-sourcing **N**oise **o**f **W**eb (NoW) dataset, a new powerful benchmark for NCL. In the future, we will explore PC$^2$'s potential in other areas of multimodal learning.

## Acknowledgements

This work is supported by the National Key R&D Program of China (2023ZD0120700, 2023ZD0120701), NSFC Project (62222604, 62206052, 62192783), State Key Laboratory Fund (ZZKT2024A14), Jiangsu Natural Science Foundation Project (BK20210224), China Postdoctoral Science Foundation (2024M750424), and Fundamental Research Funds for the Central Universities (020214380120).

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
