# OpenReview forum: "PC$^2$: Pseudo-Classification Based Pseudo-Captioning for Noisy Correspondence Learning in Cross-Modal Retrieval"
_acmmm.org/ACMMM/2024/Conference — MM2024 Poster_

### Official Review · Reviewer_dHDn · 2024-05-15

**Rating:** 3
**Confidence:** 4

**Summary:**

This paper proposed a new method to solve the noisy correspondence problem, named $\text{PC}^2$, which inherits a variety of strategies that are beneficial to solving NC, such as Pseudo-Classification, Pseudo-captions, and Oscillation. From the ablation experiments, each module used contributes. And, the authors proposed a new benchmark, which will make a certain contribution to the community.

**Strengths:**

1. The method is simple and clear.
2. Experiments prove that the method is effective.
3. Writing meets standards.

**Limitations:**

1. I think innovation may be limited. Among them, Pseudo-Classification is very similar to ID loss technology (Instance loss) in Reid. For example: Dual-path convolutional image-text embeddings with instance loss, and Cross-Modal Implicit Relation Reasoning Aligning for Text-to-Image Person Retrieval.
2. Pseudo-subtitles Logically speaking, the larger the mini-batch, the greater the probability that the retrieval may be correct, and the performance should be higher. However, the results presented in Table 5 are opposite.
3. The performance of the newly proposed benchmark is very low. Has the author considered refining the test set? Because the NC problem seems to be specific to the training set.
4. The author mentioned too many places and asked me to find supplementary materials, which greatly affected my reading.
5. The comparison is not completely SOTA and lacks extensive citations of recent related works on noisy correspondence.

**Suitability:**

3

---

### Official Review · Reviewer_L6hW · 2024-05-16

**Rating:** 3
**Confidence:** 3

**Summary:**

This paper proposes a Pseudo-Classification based Pseudo-Captioning framework to address the noisy correspondence challenge. The authors establish a pseudo-classification task to provide more information and tangible supervision for each mismatched pair. Building upon this, the oscillations of pseudo-classification are used to assist the correction of correspondence. Moreover, the authors constructed a new dataset named Noise of Web (NoW) to simulate real-world scenarios better. The proposed method exhibits significant performance improvement compared to state-of-the-art baseline methods on the NOW dataset.

**Strengths:**

1: This paper leverages the pseudo-classification capability of the model to provide more usable information for samples, offering more intuitive supervision for mismatched pairs.

2: The model uses the correspondence correction mechanism, to perform effective correspondence correction for clean data.

3: This paper constructs a new dataset that better simulates the noisy correspondence in the real world, providing a basis for future research to explore new methods and solve new problems.

**Limitations:**

1: Could the authors elaborate on how the curve parameter m in Eq. 5 is defined?

2: In Section 4.4, can the correspondence correction mechanism be applied to the re-matched dataset for further refinement?

3: The authors did not conduct the parametric sensitivity analysis for the parameters in Eq.10.

4: Does mini-batch training introduce errors in global matching?

5: In Section 3.2, the authors mentioned that the noise ratio in the dataset approaches 70%. However, they did not validate its performance under extreme conditions.

6: The comparison methods are not consistent across this paper. 7 methods are compared at 0% noise, 8 methods are compared at 50% noise, and 9 methods are compared for other scenarios.

7: The experiments are insufficient, lacking visualization results, the compared methods are outdated, and the argumentation is not persuasive.

**Suitability:**

3

---

### Official Review · Reviewer_JZqC · 2024-05-24

**Rating:** 4
**Confidence:** 4

**Summary:**

In this paper, the authors present a noise correspondence learning framework, which is a threefold strategy. It first establishes an auxiliary pseudo-classification task to interpret captions as categorical labels. Thereafter, the proposed method generates pseudo-captions to provide more information for mismatched pairs. Finally, it adopts the oscillation of pseudo-classification to assistant the correction of correspondence. The authors demonstrate the effectiveness of the proposed method by experiments. In addition, the authors also develop a dataset for noise correspondence learning.

**Strengths:**

1.	The idea of this work is technically sound. This paper is well organized.
2.	Extensive experiments are conducted. And the proposed method achieves competitive results.
3.	A new dataset is developed.

**Limitations:**

1.	Some sentences are confusing. For example, in 5.3 Results and analysis, the authors state that “SoC” shows promising results on……”. What is SoC?
2.	There are many writing problems, e.g.. grammars.

**Suitability:**

3

---

### Official Review · Reviewer_sa22 · 2024-05-25

**Rating:** 4
**Confidence:** 3

**Summary:**

This paper proposes a cross-modal retrieval method targeting noisy correspondence learning. The method initially employs an auxiliary task termed "Pseudo-classification," where captions are used as classification labels to guide the model in learning the semantic relationships between image-text pairs. Subsequently, leveraging its classification capabilities, the method constructs pseudo-captions for mismatched pairs, thereby providing richer supervision. Finally, the oscillation of pseudo-classification is utilized to assist in the correction of correspondence. Additionally, a more complex and natural dataset, Noise of Web, has been constructed to train and evaluate the model's performance.

**Strengths:**

- The paper introduces a more complex and natural dataset for training and evaluating cross-modal retrieval models.
- The paper employs an auxiliary "pseudo-classification" task to learn the semantic relationships between image-text pairs, a novel use of pseudo-captions for richer supervision of mismatched pairs, and a correspondence correction mechanism. This approach demonstrates significant innovation.
- The paper is very well written and easily understood. The organization is also clear.
- This paper conducts a comparative analysis with existing methods and proves the superiority of the proposed method in cross-moal retrieval task.

**Limitations:**

- In Section 4.1, setting the threshold at 0.5 may be overly simplistic and may not accurately distinguish between clean and noisy data. There are no experimental results provided to validate this setting.
- How are captions used as classification labels? Are different captions used directly as labels? Could there be cases where captions have the same semantics or are missing?
- In Section 5.3, what does SoC represent?

**Suitability:**

3

---

### Meta-Review · Area_Chair_To3h · 2024-07-02

**Recommendation:** Accept (Poster)
**Confidence:** 5

**Metareview:**

All four reviewers agree that the authors addressed their concerns in the rebuttal. Authors need to implement reviewers' feedback for the final version.